# Anti-Aging Effects of Nanovesicles Derived from Human Tonsil-Derived Mesenchymal Stem Cells

**Dohyun Kim [1,†], Youngdae Lee [2,†], Kwangsook Park [2], Danbi Park [1], Won Jai Lee [2], Tai Suk Roh [2], Hyungju Cho [3,*] and Wooyeol Baek [1,2,*]**

1   Plcoskin Inc., Seoul 120-752, Korea; dohyun3343@plcoskin.com (D.K.); plzi501@gmail.com (D.P.)
2   Department of Plastic & Reconstructive Surgery, Institute for Human Tissue Restoration, Severance Hospital, Yonsei University College of Medicine, Seoul 03722, Korea; mdlyd@daum.net (Y.L.); soooook1606@yuhs.ac (K.P.); pswjlee@yuhs.ac (W.J.L.); rohts@yuhs.ac (T.S.R.)
3   Department of Otorhinolaryngology, Yonsei University College of Medicine, 50 Yonsei-ro, Seodaemun-gu, Seoul 120-752, Korea
*   Correspondence: hyungjucho@yuhs.ac (H.C.); parande@yuhs.ac (W.B.)
†   Equally contribution.

**Abstract:** Growing evidence has demonstrated that biomimetic nanovesicles produced from specific cells show bioactive properties such as anti-tumor or anti-inflammatory activities. However, the properties of these nanovesicles are very diverse, depending on their cell sources. In this study, human tonsil-derived mesenchymal stem cells (TMSCs) were used in the production of functional biomimetic nanovesicles with anti-senescence. TMSCs were isolated from human tonsil tissue obtained by tonsillectomy. TMSC-derived nanovesicles (TMSC-NVs) were produced by serial extrusion using a mini-extruder. Western blotting and particle analysis were performed for characterization of TMSC-NVs. They were applied to both replicative and ultraviolet B-induced senescent human dermal fibroblasts in vitro. Following six days of treatment, analysis of the proliferation and senescence level of fibroblasts was performed using cell counting and senescence-associated β-galactosidase assay, respectively. Treatment with TMSC-NVs enhanced the cell proliferation and reduced the activity of senescence-associated β-galactosidase in both replicative and ultraviolet B-induced senescent cells. Treatment with TMSC-NVs resulted in increased expression of extracellular matrix and anti-oxidant genes. Treatment with TMSC-NVs resulted in reduced expression of vinculin in focal adhesion. These results show that TMSC-NVs have an effect on recovering from cellular senescence by oxidative stress and can be applied as useful materials for the development of skin rejuvenation.

**Keywords:** human tonsil-derived mesenchymal stem cells; biomimetic nanovesicle; skin rejuvenation





## 1. Introduction

Aging is characterized by time-dependent loss of function and regenerative properties of organisms. Factors related to skin aging cause damage to cells, delaying skin regeneration and cell proliferation, known as cellular senescence. Cellular senescence is characterized by an irreversible arrest of the cell cycle [1] and alteration of the focal adhesive cytoskeleton [2,3]. Cellular senescence in skin tissues is induced by various factors, such as oxidative stress [4,5], mitochondrial dysfunction [6,7], and ultraviolet irradiation [8,9]. For the past several decades, many researchers have been conducting research to overcome cellular senescence for skin regeneration.

Recently, to overcome cellular senescence, many studies have focused on the tissue regenerative potential of exosomes. Exosomes, which are known as nano-size biomimetic nanovesicles secreted across the plasma membrane from the originated cells by the endocytic pathway [10,11], contain several components, including miRNA, mRNA, and proteins [12–14]. In addition, exosomes have been studied for skin rejuvenation and anti-aging approaches. For example, Choi et al. demonstrated that exosomes derived from

adipose-derived mesenchymal stem cells promote the proliferation and migration of human dermal fibroblasts (HDFs) [15]. Kim at el. demonstrated that the exosomes secreted from induced pluripotent stem cell-derived mesenchymal stem cells increase the proliferation of HDFs and HaCaT (immortalized human keratinocytes) [16]. Human umbilical cord blood stem cell-derived exosomes were studied for cutaneous wound healing via stimulation of the Wnt signaling pathway [17] and skin rejuvenation [18]. However, despite the potential of exosomes for therapeutic purposes, there are several hurdles, including low efficiency, long procedure time, and high technical expertise [19].

To overcome those hurdles, many researchers have focused on the direct production of exosome-mimetic nanovesicles from somatic cells [20–23]. These biomimetic nanovesicles can be directly isolated from the desired cells by sonication and/or extrusion and are reported to share similar characteristics with exosomes. Given the similar properties, cell-derived biomimetic nanovesicles could be utilized for drug delivery [24,25], tissue regeneration [26,27], and cancer targeting [28,29]. In particular, it was reported that nanovesicles derived from human tonsil-derived mesenchymal stem cells attenuate liver fibrosis and inflammation [30,31]. However, the anti-aging effects of nanovesicles derived from human tonsil-derived mesenchymal stem cells (TMSCs) have not yet been investigated.

In this study, we examined the question of whether the nanovesicles derived from human tonsil-derived mesenchymal stem cells (TMSC-NVs) have a promising potential to decrease cellular senescence and increase cell proliferation. Both intrinsic replicative senescence and extrinsic ultraviolet B (UV)-irradiated senescence were employed in an in vitro model. TMSC-NVs were applied to these senescence models for assessment of their anti-aging effect.

## 2. Materials and Methods

### 2.1. Isolation of Tonsil-Derived Mesenchymal Stem Cells

TMSCs were isolated from human tonsil tissue obtained by tonsillectomy. Human tonsil tissue was washed with phosphate-buffered saline (PBS) (Welgene, Seoul, South Korea) containing 2% antibiotics–antimycotics (Thermo Fisher Scientific, Waltham, MA, USA). Tissue was chopped and digested using 210 U/mL of collagenase type 1 (Gibco, New York, NY, USA) and 4 KU/mL of DNase 1 (Sigma, St. Louis, MO, USA) in Dulbecco's modified Eagle's medium (DMEM) in low glucose (Gibco, New York, NY, USA) at 37 °C for 1 h 30 min. The digested product was filtered through a 40 μm strainer and centrifuged at 1300 rpm for 3 min. The pellet was washed twice with fresh DMEM. Cells were cultured in DMEM containing 10% fetal bovine serum (Gibco, New York, NY, USA) and 1% antibiotics–antimycotics at 37 °C and 5% $CO_2$ atmosphere. The media was changed every two days. Flow cytometry with anti-CD90, anti-CD105, and anti-CD73 antibodies (Biolegend, San Diego, CA, USA) was used for characterization of TMSCs.

### 2.2. Production of Nanovesicles Derived from TMSCs

For the manufacture of the TMSC-derived nanovesicles (TMSC-NVs), TMSCs at passages 2–4 were dissociated with TrypLE express solution (Gibco, New York, NY, USA) at 37 °C for 2 min. TMSCs were suspended with culture media and centrifuged at 1300 rpm for 3 min. The cell pellet was washed twice with PBS and cells were resuspended at a density of $1 \times 10^6$ cells/mL in PBS. Cells were sequentially passed through filter papers with pore sizes of 10, 5, and 0.4 um using a mini extruder (Avanti polar lipids, Alabaster, AL, USA) to produce cell-derived nanovesicles. The size and shape of TMSC-NVs were determined by dynamic light scattering (DLS) and transmission electron microscopy (TEM), respectively. The concentration of TMSC-NVs was measured using Micro BCA™ Protein Assay Kit (Thermo Fisher Scientific, Waltham, MA, USA) and the senescent cells were treated with 50 μg/mL of TMSC-NVs only once.

### 2.3. Western Blot

To determine the protein expression of the nanovesicles, cells and nanovesicles were harvested and lysed in RIPA buffer (Sigma, St. Louis, MO, USA). Lysates were centrifuged at 13,000 rpm for 20 min for removal of cellular debris. The amount of protein in the supernatant was measured using a Micro BCA™ Protein Assay Kit. An amount of 20 µg of total protein was loaded and separated on a 10% SDS-PAGE gel. After loading, separated protein was transferred onto a membrane, which was blocked with 5% BSA solution for 30 min. For immunoblotting, rabbit anti-CD9 (1:2000), anti-CD63 (1:2000), and anti-beta actin (1:5000) primary antibodies (Abcam, Cambridge, UK) were applied at 4 °C overnight. For chemiluminescence detection of proteins, HRP-conjugated goat anti-rabbit IgG (H + L) (1:5000) secondary antibody (Invitrogen, Carlsbad, CA, USA) was applied at room temperature for 2 h and Amersham™ ECL Select™ (Thermo Fisher Scientific, Waltham, MA, USA) was used for detection.

### 2.4. Cell Culture

HDFs were purchased from American Type Culture Collection (ATCC, Manassas, VA, USA). Cells were cultured in DMEM including 10% fetal bovine serum and 1% antibiotics–antimycotics at 37 °C and 5% $CO_2$ atmosphere. The media was changed every two days. Intrinsic replicative senescent cells were produced by repeated passages. Cells with passage 3 and 15 were identified as 'young' and 'old' states, respectively. Extrinsic senescent cells were treated by UV irradiation with 200 mJ/$cm^2$ of UV.

### 2.5. Senescence-Associated β-Galactosidase Assay (SA-β-Galactosidase Assay)

As described by Kurz et al., [32], senescence-associated beta-galactosidase is an enzyme to catalyze the hydrolysis of β-galactosides into monosaccharides. It is detectable in senescent cells and tissues at only pH 6.0, not at pH 4.0. As a biomarker of cellular senescence, its activity can be detected using a chromogenic assay using 5-bromo-4-chloro-3-indoyl β-D-galactopyranoside (X-gal), which converts to an insoluble blue compound. Here, SA-β-galactosidase assay was performed using a cellular senescence staining kit (Cell Biolabs, San Diego, CA, USA). All cells were fixed with 0.25% glutaraldehyde solution at room temperature for 5 min. After discarding the fixation solution, each well was gently washed with fresh PBS three times. Cell staining working solution was added, followed by incubation of the cells overnight at 37 °C as described. After reaction of the working solution, supernatant was discarded and cells were gently washed with PBS two times. Quantitative data were measured by documentation of colorization ratio of the senescent cells.

### 2.6. Immunocytochemistry

To perform the immunocytochemistry, cells were fixed with 4% paraformaldehyde for 30 min. Cells were permeabilized with 0.05% Triton X-100 (Sigma, St. Louis, MO, USA) for 15 min. After permeabilization, cells were blocked with 1% bovine serum albumin (BSA) (Sigma, St. Louis, MO, USA) at room temperature for 30 min, followed by incubation with anti-vinculin (1:200) at room temperature for 1 h. After washing with PBS, goat anti-rabbit IgG H&L Alexa Fluor 488 secondary antibodies and tetramethylrhodamine-conjugated phalloidin (1:200) were applied for 1 h in the dark. To confirm ECM production, cells were fixed, blocked, and incubated with anti-collagen 1 primary antibody (1:200) for 1 h and goat anti-rabbit IgG H&L Alexa Fluor 488 (1:200). All antibodies used for immunocytochemistry were purchased from Abcam (Cambridge, MA, USA). Immunocytochemistry was counterstained with DAPI nuclear staining and examined under a ZEISS LSM700 confocal microscope (Zeiss, Oberkochen, Germany).

### 2.7. Quantitative Real-Time Polymerase Chain Reaction (qPCR)

To confirm the RNA expression of the cells, total RNA was extracted using Trizol reagent (Ambion, Waltham, MA, USA). Six days after treatment with TMSC-NV, 500 µL of

Trizol reagent was applied to lyse the cells. Reagent was treated with 100 μL of chloroform (Sigma, St. Louis, MO, USA) and incubated on ice for 15 min. After incubation samples were centrifuged at 13,000 rpm for 15 min and transparent supernatant was obtained. The supernatant was treated with the same volume of isopropanol (Sigma, St. Louis, MO, USA), incubated at room temperature for 15 min, and centrifuged at 1300 rpm for 15 min for precipitation of RNA. Concentration of total RNA was measured using NanoDrop 2000 (Thermo Fisher Scientific, Waltham, MA, USA) and 1 μg of total RNA was used for cDNA synthesis with the SuperScript™ III First-Strand Synthesis System (Invitrogen, Carlsbad, CA, USA) according to the described procedure. Power SYBR™ Green PCR Master Mix (Applied Biosystems, San Francisco, CA, USA) was used in performance of qPCR and the reaction was measured using the StepOnePlus real-time PCR system (Applied Biosystems, San Francisco, CA, USA).

### 2.8. Statistical Analysis

All experiments in this study were repeated at least three times. All graphs in this study were drawn using GraphPad Prism 5 software (GraphPad Software Inc., San Diego, CA, USA) and are presented as mean ± SEM. Significant differences were determined using one-way ANOVA analysis (* $p < 0.05$, ** $p < 0.01$, *** $p < 0.001$).

## 3. Results

### 3.1. Production of Nanovesicles Derived from TMSC-NVs

TMSCs were isolated and cultured from human tonsil tissues followed by tonsillectomy. TMSCs have a fibroblastic morphology (Figure 1a). Flow cytometry data showed that TMSCs are positive for typical MSC markers, including CD90, CD105, and CD73 (Figure 1b). After fabrication of TMSC-NVs, Western blot, TEM, and DLS were performed for characterization of TMSC-NVs (Figure 1c,d). TMSC-NVs express exosome markers such as CD9 and CD63. TMSC-NVs have a spherical morphology and the diameters of TMSC-NVs show as two peaks (88.5 and 228.3 nm). These results show that the characteristics of TMSC-NVs are similar to those of exosomes.

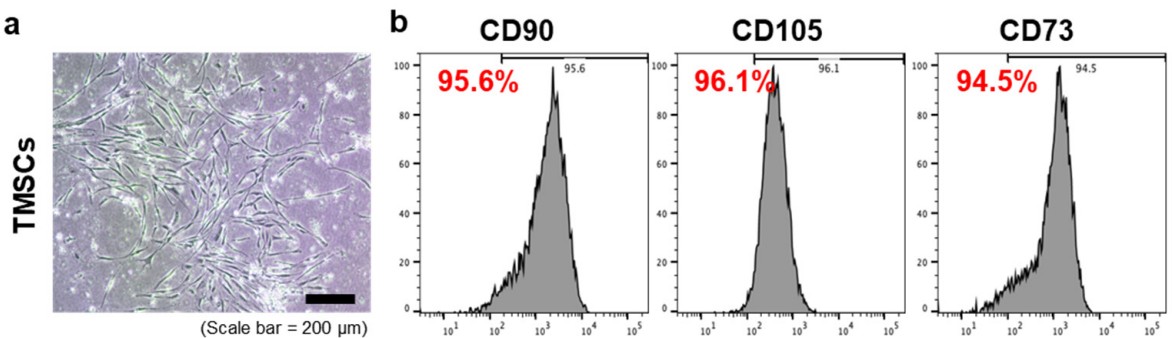

**Figure 1.** *Cont.*

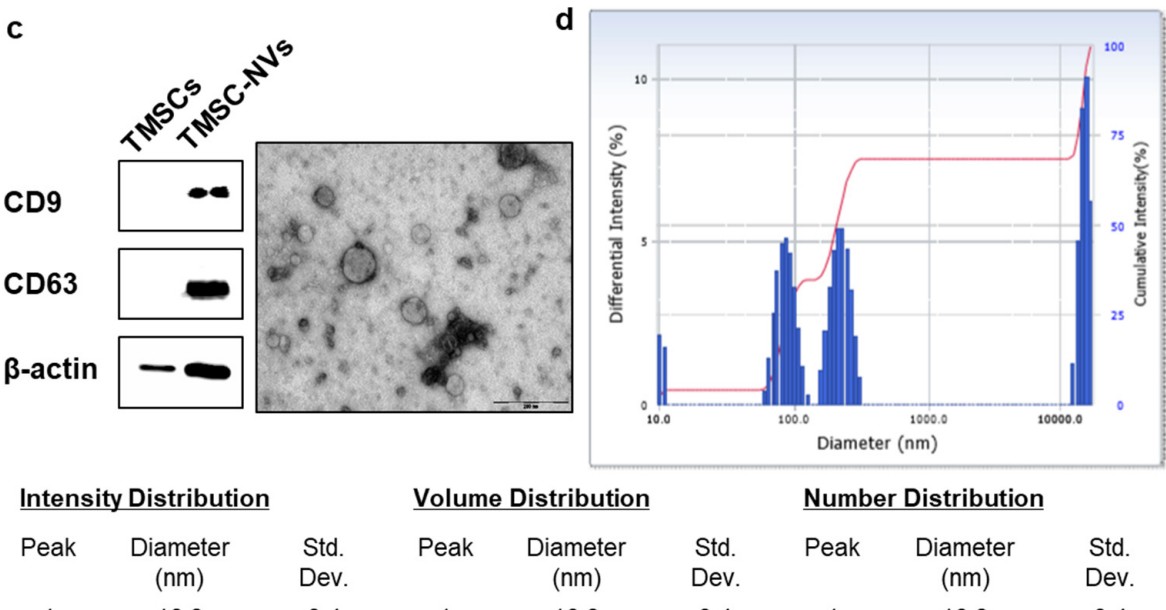

**Figure 1.** Production of nanovesicles derived from tonsil-derived mesenchymal stem cells (TMSC-NVs). (**a**) Morphology of human tonsil-derived mesenchymal stem cells, scale bar = 200 μm. (**b**) Expression of surface markers of human tonsil-derived mesenchymal stem cells. (**c**) Protein expression and SEM image of TMSC-NVs. Protein level was normalized by β-actin. (**d**) Analysis of dynamic light scattering of TMSC-NVs.

### 3.2. Anti-Aging Effect of TMSC-NVs in the Passage-Associated Senescence Model

To confirm the anti-aging properties, TMSC-NVs were added to the passage-associated senescence model and cell morphology and proliferation was observed (Figure 2a,b). As shown in Figure 2a,b, treatment with TMSC-NVs increased the cell proliferation of old HDFs. We performed a SA-β-galactosidase assay to confirm the anti-aging role of TMSC-NVs (Figure 2c). The results show that treatment with TMSC-NVs decreased β-galactosidase activity of old HDFs. Quantitative data from the SA-β-galactosidase assay showed that the ratio of senescent cells was reduced by treatment with TMSC-NVs, which supported that treatment with TMSC-NVs decreased the senescence level of HDFs (Figure 2d).

Additionally, the protein expression of vinculin in focal adhesion and the morphological change in the actin cytoskeleton followed by treatment with TMSC-NVs was examined (Figure 2e,f). The results of the immunofluorescence analysis show increased commitment of vinculin in focal adhesion in old HDFs, which was decreased by treatment with TMSC-NVs. These results show that TMSC-NVs increase the proliferation of HDFs and decrease the senescence induced by passages.

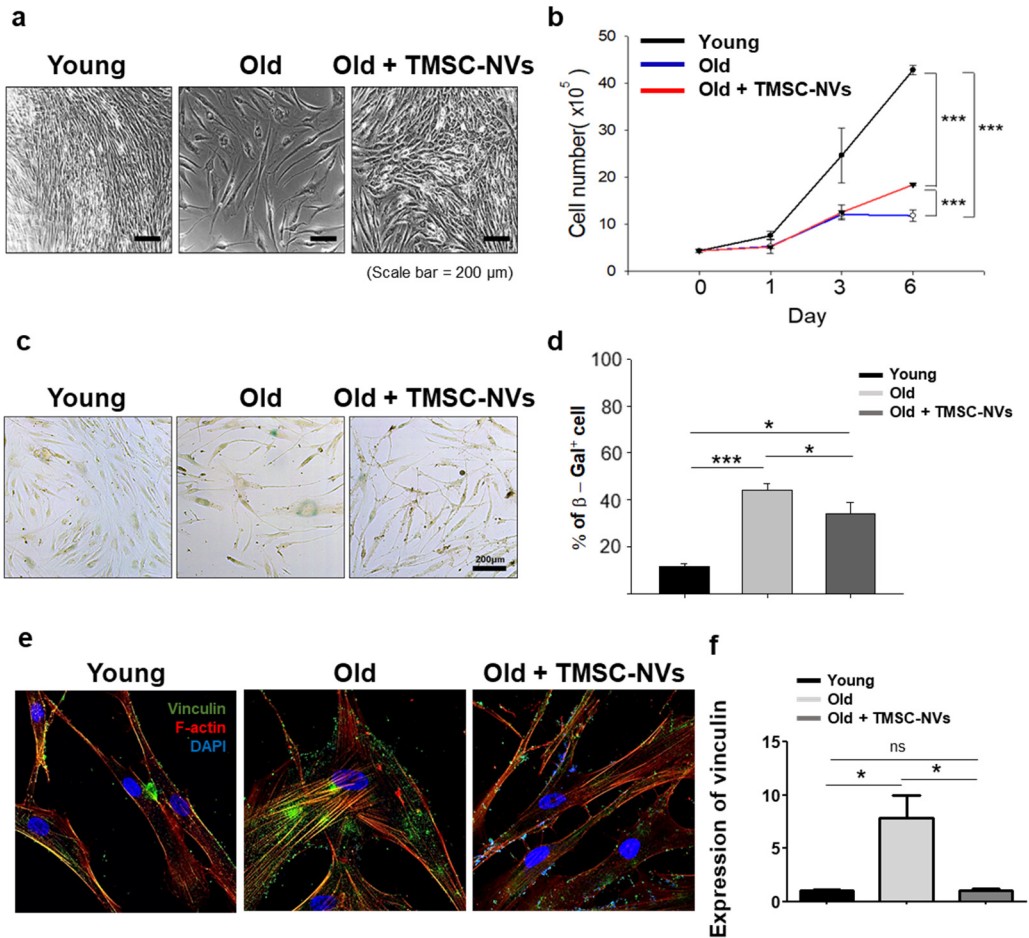

**Figure 2.** Regulation of proliferation and senescence by treatment with TMSC-NVs in the passage-associated senescence model. (**a**) Morphological change in HDFs. Scale bar = 200 μm. (**b**) Proliferation test of passage-associated senescent HDFs after treatment with TMSC-NVs. (**c**) Senescence-associated (SA) β-galactosidase assay, scale bar = 200 μm. (**d**) Quantitative analysis of the SA-β-galactosidase assay. (**e**) Expression of vinculin in focal adhesion of HDFs. (**f**) Quantitative data of vinculin expression in focal adhesion. Significant differences among groups were determined by one-way ANOVA (ns > 0.05, * $p < 0.05$, ** $p < 0.01$, *** $p < 0.001$).

### 3.3. Regulation of Extracellular Matrix and Anti-Oxidant Gene by TMSC-NVs in the Passage-Associated Senescence Model

To confirm the anti-aging properties of TMSC-NVs in terms of molecular biology, the gene expression of the extracellular matrix (ECM) production and senescence-related anti-oxidant gene followed by TMSC-NVs treatment was examined (Figure 3a). The mRNA level of collagen type 1 (*COL1*) and *ELASTIN*, which is decreased in old HDFs compared to young HDFs, showed that treatment with TMSC-NVs resulted in upregulated ECM production. Similarly, the mRNA expression of *SOD2* and *HMOX1*, the anti-oxidant gene, was increased by treatment with TMSC-NV in old HDFs. In addition, the protein expression of COL1 examined by immunofluorescence showed that treatment with TMSC-NVs resulted in significantly increased ECM production in the passage-associated senescent HDFs (Figure 3b,c). According to these results, treatment with TMSC-NVs resulted in recovery of the ECM production and senescence-reducible anti-oxidant gene in senescent cells.

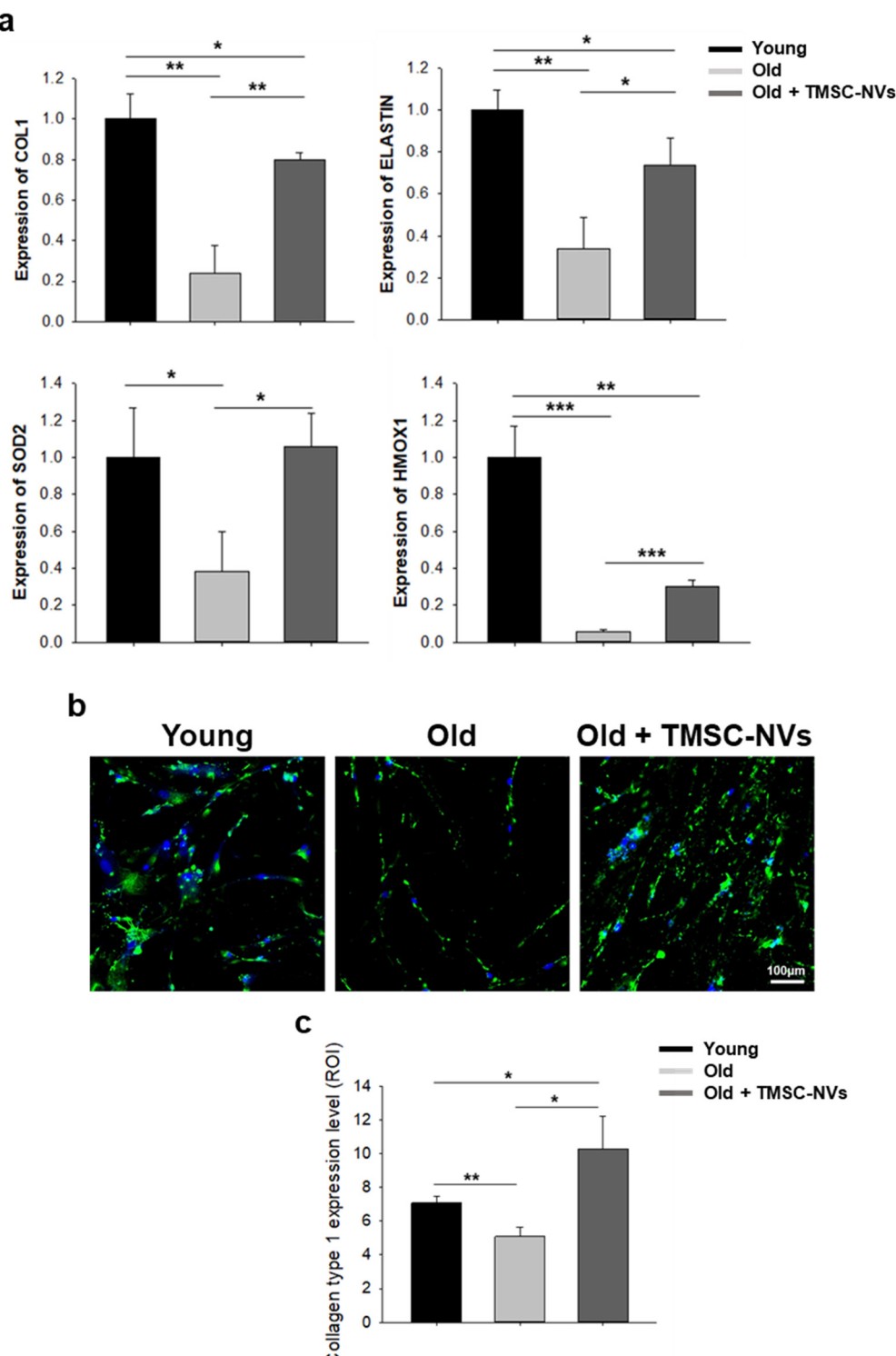

**Figure 3.** Regulation of extracellular matrix and anti-oxidant gene by treatment with TMSC-NVs in the passage-associated senescence model. (**a**) mRNA expression of *COL1, ELASTIN, SOD2,* and *HMOX1* in passage-associated senescent HDFs after treatment with TMSC-NVs. (**b**) Immunofluorescence analysis of collagen type 1 of passage-associated senescent HDFs after treatment with TMSC-NVs. (**c**) Quantitative data of immunofluorescence analysis. Significant differences among groups were determined by one-way ANOVA (* $p < 0.05$, ** $p < 0.01$, *** $p < 0.001$).

### 3.4. Anti-Aging Effect of TMSC-NV in the UV-Induced Senescence Model

Based on the results of the intrinsic senescence model, TMSC-NVs were added to the extrinsic senescence model, the UV-induced senescence model. As a result, proliferation of HDFs was decreased after UV irradiation and increased by treatment with TMSC-NVs, compared with UV-induced senescent HDFs (Figure 4a,b). In addition, the result of the SA-β-galactosidase assay shows that TMSC-NVs decreased the β-galactosidase activity of UV-induced senescent cells (Figure 4c) and the quantitative data from the SA-β-galactosidase assay support that UV-induced senescence was significantly decreased after treatment with TMSC-NVs, similar to the passage-associated senescence model (Figure 4d).

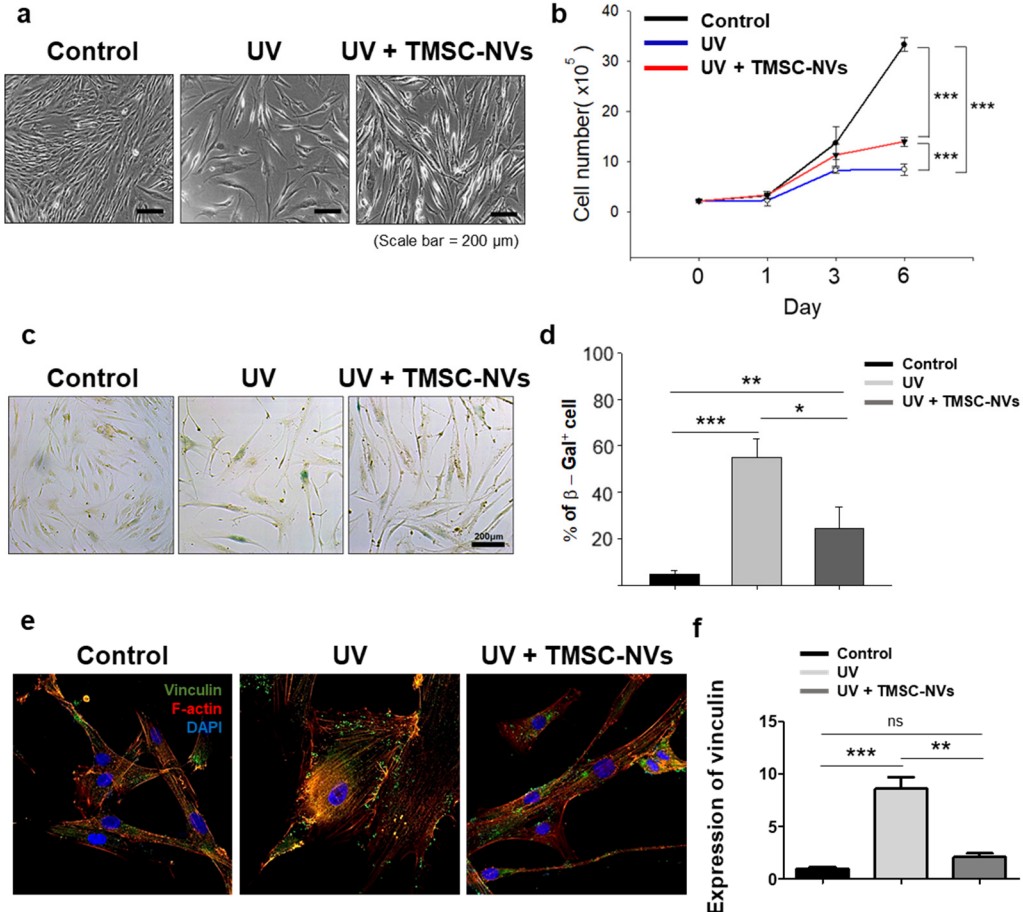

**Figure 4.** Regulation of proliferation and senescence by treatment with TMSC-NVs in the UV-induced senescence model. (**a**) Morphological change in HDFs by treatment. Scale bar = 200 μm. (**b**) Proliferation test of UV-induced senescent HDFs after treatment with TMSC-NVs. (**c**) SA-β-galactosidase assay of UV-induced senescent HDFs after treatment with TMSC-NVs, scale bar = 200 μm. (**d**) Quantitative analysis of the SA-β-galactosidase assay. (**e**) Expression of vinculin in focal adhesion of HDFs. (**f**) Quantitative data of vinculin expression in focal adhesion. Significant differences among group were determined by one-way ANOVA (ns; not significant, * $p < 0.05$, ** $p < 0.01$, *** $p < 0.001$).

Additionally, vinculin expression in the focal adhesion and arrangement of the cytoskeleton of HDFs was examined through treatment with TMSC-NVs (Figure 4e,f). The results of the immunofluorescence analysis show increased expression of vinculin in the UV-induced senescent HDFs and treatment with TMSC-NVs resulted in decreased expression of vinculin in focal adhesion. These results show that TMSC-NVs increase the proliferation of HDFs and decrease the cellular senescence in the UV-induced senescence model.

### 3.5. Regulation of Extracellular Matrix and Anti-Oxidant Gene by Treatment with TMSC-NV in the UV-Induced Senescence Model

To confirm the anti-aging properties of TMSC-NVs in a UV-induced senescence model in terms of molecular biology, mRNA expression of senescence-related ECM production and anti-oxidant gene was examined by qPCR. The results of qPCR show that *COL1* and *ELASTIN* were decreased after UV irradiation and treatment with TMSC-NVs resulted in significantly increased *COL1*; however, *ELASTIN* was not increased. The anti-oxidant genes, *SOD2* and *HMOX1*, were decreased by UV irradiation and increased by treatment with TMSC-NVs (Figure 5a). In addition, the results of immunofluorescence analysis show decreased expression of collagen type 1 in UV-induced senescent HDFs, which was increased by treatment with TMSC-NVs (Figure 5b,c). These results show that TMSC-NVs increase the ECM production and senescence-reducible anti-oxidant gene.

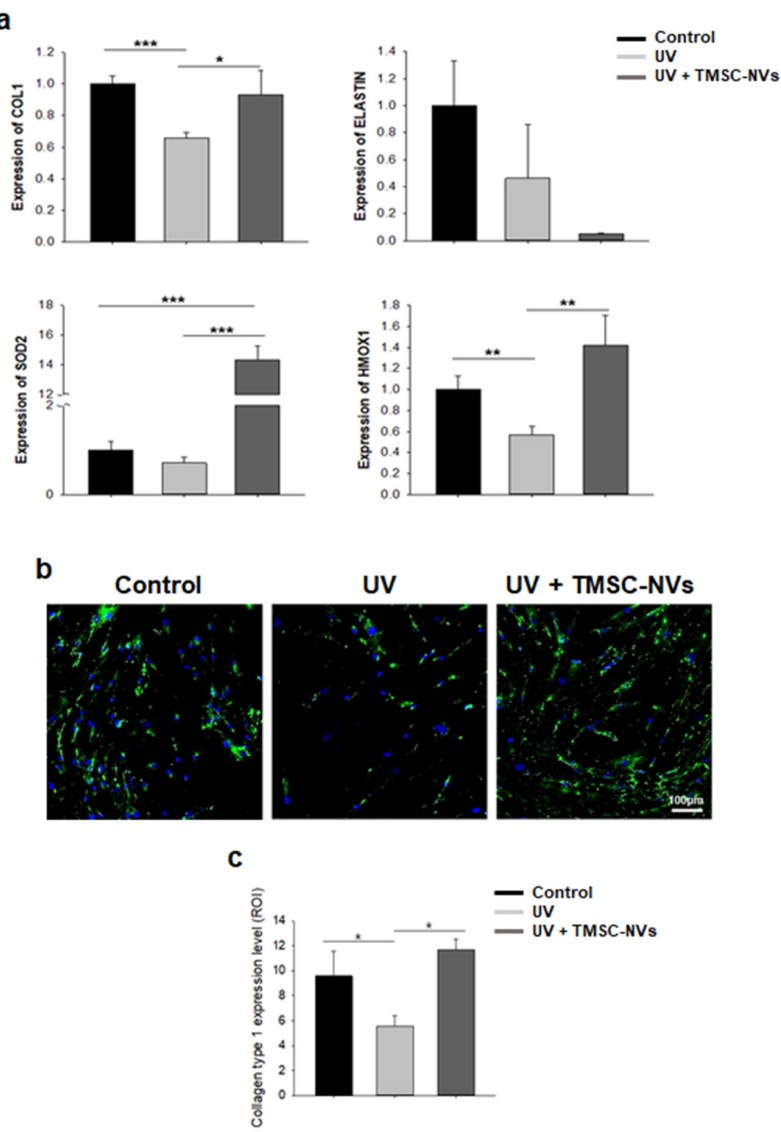

**Figure 5.** Regulation of extracellular matrix and anti-oxidant gene by treatment with TMSC-NVs in the UV-induced senescence model. (**a**) mRNA expression of *COL1, ELASTIN, SOD2,* and *HMOX1* in UV-induced senescent HDFs after treatment with TMSC-NVs. (**b**) Immunofluorescence analysis of collagen type 1 of UV-induced senescent HDFs after treatment with TMSC-NVs. (**c**) Quantitative data of immunofluorescence analysis. Significant differences among groups were determined by one-way ANOVA (* $p < 0.05$, ** $p < 0.01$, *** $p < 0.001$).

## 4. Discussion

In this study, we explored the question of whether the nanovesicles derived from TMSCs can exert anti-aging properties. We isolated and confirmed characteristics of TMSCs and TMSC-NVs. TMSCs were highly proliferative MSC-like cells and TMSC-NVs possessed characteristics similar to those of exosomes. TMSC-NVs accelerated the proliferation and reduced senescence-associated β-galactosidase activities and vinculin expression in focal adhesion of both senescent HDFs. ECM production and the anti-oxidant gene involved in cellular senescence were upregulated in the senescent HDFs by treatment with TMSC-NVs. With these results, we suggest that TMSC-NVs can be utilized for skin rejuvenation and anti-aging purposes.

In the past decade, the poor yield and inefficient separation procedure for exosome production have been challenged [19]. To overcome these issues, we sought to produce exosome-mimetic nanovesicles from human tonsil-derived mesenchymal stem cells using a relatively simple extrusion procedure [33]. In this study, even though the comparative data of characteristics with exosomes are not shown, TMSC-NVs express the exosome-specific markers (CD9 and CD63) and the size of TMSC-NVs is similar to that of exosomes, meaning that the characteristics of TMSC-NVs are similar to those of exosomes, and TMSC-NVs can be utilized as an alternative for exosome therapies.

It has been reported that the mRNA and miRNA profiles of exosomes differ from their originated cells [13]. Here, numerous pieces of research have revealed the specific marker in exosomes to demonstrate the mechanism of tissue regeneration mediated by exosomes. For instance, Ying et al. demonstrated that exosome-mediated delivery of miR-155 regulates insulin sensitivity and glucose homeostasis [34]. Xin et al. reported that mesenchymal stem cell-derived exosomes are transferred to neurons and astrocytes and exosome-mediated miR-133 plays a key role in neurological recovery from stroke [35]. Even though the presented results show that the TMSC-NVs can be utilized as an alternative to exosomes for skin rejuvenation, the mechanism of regulation of cellular senescence via treatment with TMSC-NVs is unclear. Therefore, in further study, we plan to discover the key marker of TMSC-NVs in the regulation of cellular senescence.

Cellular senescence is characterized as an irreversible arrest of cell growth, which occurs through alternation of the focal adhesive cytoskeleton [2,3]. In the present study, cellular senescence was induced by both passage-associated senescent HDFs and UV-induced senescent HDFs, and TMSC-NVs decreased cellular senescence and increased cell proliferation, anti-oxidant gene expression and extracellular matrix production. However, the expression of *ELASTIN* was increased in the passage-associated senescent HDFs but not in the UV-induced senescent HDFs. Our prediction is a difference in the senescence induction mechanism of both senescence models. It has been reported that senescence can be induced by various factors and the pathway of senescence induction can be different in each stimuli [36]. In particular, Pascal et al. reported different mRNA profiles among representative cellular senescence models, including a replicative senescence model, tert-butyl hydroperoxide-induced senescence model, and EtOH-induced senescence model [37]. Given these reports, we plan to explore the role of senescence in the regulation of extracellular matrix expression in each modeling procedure, which may provide further research for skin rejuvenation.

The next frontier of exosome-mimetic nanovesicles for skin rejuvenation is demonstration of the regenerative potential from ex vivo to clinical trials. In this study, we determined the proangiogenic and anti-inflammatory effects of extracellular vesicles, which will be helpful in regenerative medicine.

## 5. Conclusions

This study demonstrated that human tonsil-derived mesenchymal stem cell-derived nanovesicles share characteristics with exosomes and increase the proliferation of senescent HDFs. The senescence-associated β-galactosidase activities and vinculin expression in senescent cells were reduced by treatment with TMSC-NVs. The gene expression of the

extracellular matrix production and anti-oxidant gene were enhanced by treatment with TMSC-NVs. These findings could contribute to the development of skin rejuvenation tools and desirable cosmetic products, once the clinical examination shows promising effectiveness.

**Author Contributions:** Conceptualization, K.P., H.C. and W.B.; methodology, D.K. and D.P.; writing— original draft preparation, D.K. and Y.L.; writing—review and editing, W.J.L., T.S.R., K.P. and W.B.; supervision, W.J.L., H.C. and W.B.; funding acquisition, T.S.R. and W.B. All authors have read and agreed to the published version of the manuscript.

**Funding:** This research was funded by the Basic Science Research Program through the National Research Foundation of Korea (NRF) funded by the Ministry of Education (NRF-2018R1D1A1B07051132) and by a faculty research grant of Yonsei University College of Medicine (6-2019-0186).

**Institutional Review Board Statement:** The study was conducted according to the guidelines of the Declaration of Helsinki, and approved by the Institutional Review Board of Yonsei University (4-2020-0934).

**Informed Consent Statement:** Informed consent was obtained from all subjects involved in the study.

**Data Availability Statement:** Details are presented within the article in the form of tables, figures, and images in results.

**Acknowledgments:** The human tonsil tissue was provided by Hyungju Cho and Dongwon Lee. This study was carried out in part in the Yonsei Advanced Imaging center in cooperation with Carl Zeiss Microscopy, Yonsei University College of Medicine.

**Conflicts of Interest:** The authors declare no conflict of interest.

## Abbreviations

| | |
|---|---|
| DMEM | Dulbecco's modified Eagle's medium |
| ECM | Extracellular matrix |
| FBS | Fetal bovine serum |
| HDFs | Human dermal fibroblasts |
| PBS | Phosphate-buffered saline |
| SA-$\beta$-galactosidase | Senescence-associated beta galactosidase |
| TMSCs | Human tonsil-derived mesenchymal stem cells |
| TMSC-NVs | Nanovesicles derived from human tonsil-derived mesenchymal stem cells |
| UV | Ultraviolet B |

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
