# Peer review of "Anti-Aging Effects of Nanovesicles Derived from Human Tonsil-Derived Mesenchymal Stem Cells"

_applsci, doi:10.3390/app11135780_

Round 1
Reviewer 1 Report
In this manuscript, Kim et al., showed that functional analysis of nanovesicles from human tonsil-derived mesenchymal stem cells. The authors demonstrated that these nanovesicles have an anti-aging effect on human dermal fibroblasts (HDFs). Many studies have focused on anti-aging and extracellular vesicles have considered one of the tools of tissue regeneration. The topic of this manuscript is hot; however, some studies have already reported function of tonsil-derived EVs (Kim J, et al., Mol Ther. 2021, Park DJ, et al., J Nanobiotechnology. 2021). There are several concerns which the authors should address before considering publication in Applied Sciences.
Comments
#1. In the line 64-65, please add the reference focused on tonsil-derived EVs (Kim J, et al., Mol Ther. 2021, Park DJ, et al., J Nanobiotechnology. 2021).
#2. In materials and methods section, the author should add details of the materials, which the authors used (supplier name, country, etc.).
#3. In the line 164-165, the authors described that “These results showed that characteristics of TMSC-NVs are similar to those of exosomes.3.2. Figures, Tables and Schemes”. Please explain the meaning of this sentence.
#4. In addition, there is no Figure 1c, 1d and 3b notations in this text. Please add it where appropriate.
#5. In Figure 1c, the expression level of β-actin was quite different between TMSC and TMSC-NV. The authors should add the amount of protein loaded for TMSC and TMSC-NV in the materials and methods section.
#6. In Figure 2a, the morphology of HDF after processing with TMSC-NVs is shown. The authors described that the morphology was changed smaller by TMSC-NV treatment, but it is not clear from this picture. Please change the picture to a clearer one. Also, it would be beneficial to show the details of the TMSC-NV treatment (amount of NVs, frequency, duration, etc.) in the method section.
#7. Figure 2e and Figure 4e show that TMSC-NV treatment decreased immunofluorescence of vinculin in focal adhesion of old HDF, especially in Figure 4d, but the difference from old HDF after TMSC-NV treatment is not clear. So please add some quantitative analysis.
Author Response
Thank you for your kind comment.
#1. In the line 64-65, please add the reference focused on tonsil-derived EVs (Kim J, et al., Mol Ther. 2021, Park DJ, et al., J Nanobiotechnology. 2021).
As you mentioned, we have added the reference (Kim J, et al., Mol Ther. 2021, Park DJ, et al., J Nanobiotechnology. 2021) in the introduction section (line 63 – 65).
#2. In materials and methods section, the author should add details of the materials, which the authors used (supplier name, country, etc.).
We have added details of the materials (company, country) in the 2. materials and methods section.
#3. In the line 164-165, the authors described that “These results showed that characteristics of TMSC-NVs are similar to those of exosomes.3.2. Figures, Tables and Schemes”. Please explain the meaning of this sentence.
We have revised the description in the line 173 as follows:
“TMSC-NVs express exosome markers such as CD9 and CD63. TMSC-NVs have a spherical morphology and the diameters of TMSC-NVs show as two peaks (88.5 nm and 228.3 nm). These results showed that characteristics of TMSC-NVs are similar to those of exosomes.”
#4. In addition, there is no Figure 1c, 1d and 3b notations in this text. Please add it where appropriate.
We have added the notations of Figure 1c, 1d and 3b (line 170, 201) as below.
“After fabrication of TMSC-NVs, western blot, TEM, and DLS were performed for characterization of TMSC-NVs (Figure 1c and 1d).”
“In addition, the protein expression of COL1 examined by immunofluorescence showed that treatment with TMSC-NVs resulted in significantly increased ECM production in the passage-associated senescent HDFs (Figure 3b and 3c).”
#5. In Figure 1c, the expression level of β-actin was quite different between TMSC and TMSC-NV. The authors should add the amount of protein loaded for TMSC and TMSC-NV in the materials and methods section.
We have described the amount of gel-loaded protein in the line 104.
“20 μg of total protein was loaded and separated on a 10% SDS-page gel.”
#6. In Figure 2a, the morphology of HDF after processing with TMSC-NVs is shown. The authors described that the morphology was changed smaller by TMSC-NV treatment, but it is not clear from this picture. Please change the picture to a clearer one. Also, it would be beneficial to show the details of the TMSC-NV treatment (amount of NVs, frequency, duration, etc.) in the method section.
We have changed the pictures of Figure 2a and 4a to the clear one as below.
Figure 2a
Figure 4a
Also, we have described the amount and frequency of TMSC-NVs treatment in the line 98 – 99.
“The concentration of TMSC-NVs was measured using Micro BCA™ Protein Assay Kit (Thermo fisher scientific, Waltham, MA, USA) and 50 μg/mL of TMSC-NVs was treated once to the senescent cells in the present study.”
#7. Figure 2e and Figure 4e show that TMSC-NV treatment decreased immunofluorescence of vinculin in focal adhesion of old HDF, especially in Figure 4d, but the difference from old HDF after TMSC-NV treatment is not clear. So please add some quantitative analysis.
We have added the quantitative data of immunofluorescence image in Figure 2f and 4f.
Figure 2f
Figure4f

Reviewer 2 Report
Paper is interesting and worth considering for publication. The Introduction and the discussion over the results of the investigations are properly presented. The descriptions of the methodology applied needs to be improved. The same concerns the presentation of the results obtained and some editorial aspects of the paper. All recommendations are provided below.
- Section 2.1.: what antibiotics have been used during washing the human tonsil tissue (line 75)?
- During the notation of carbon dioxide, the subscript should be used (e.g. line 80).
- Section 2.5.: the main principle of SA-β-galactosidase assay should be provided.
- It is suggested to add separate subsection containing the explanations of all applied abbreviations.
- Paper should be written in a third person, not in a first one.
- Quality of Figures should be improved. Furthermore, some figures should be enlarged because they are poorly legible (e.g. Figure 1b or 1d etc.).
- From editorial viewpoint, references in brackets should be placed at the end of the sentence before the dot – this should be corrected through the whole article.
- Final conclusions should be improved to be more specified and understandable. Now, this section consists of two sentences wherein the first is very, very long. It is better to list the highlights from the investigations in few, specified sentences.
- Section References should be improved to be consistent. Now, some references contain the whole journal title and in the case of some of them their abbreviations have been given.
- Article should be re-checked grammatically.
Author Response
Thank you for your kind comment.
- Section 2.1.: what antibiotics have been used during washing the human tonsil tissue (line 75)?
We used Antibiotics-Antimycotics (Cat. No. 15240062, Thermo Fisher Scientific) in the present study. We have described the information of antibiotics in the line 78.
“Human TMSCs were isolated from tonsil tissue obtained by tonsillectomy. Human tonsil tissue was washed with phosphate-buffered saline (PBS) (Welgene, Seoul, South Korea) containing 2% antibiotics-antimycotics (Thermo fisher scientific, Waltham, MA, USA).”
- During the notation of carbon dioxide, the subscript should be used (e.g. line 80).
We have described the carbon dioxide as CO2 using subscript in the line 85, 116.
“Cells were cultured in DMEM containing 10% fetal bovine serum (Gibco, NY, USA) and 1% antibiotics-antimycotics at 37 °C and 5% CO2 atmosphere.”
- Section 2.5.: the main principle of SA-β-galactosidase assay should be provided.
We have described the principle and reference (Kurz et al., 2000) of the principle of SA-β-galactosidase assay in the line 121-126. Senescence-associated beta-galactosidase is an enzyme to catalyze the hydrolysis of β-galactosides into monosaccharides. It is detectable in senescent cells and tissues at only pH 6.0, not at pH 4.0. As a biomarker of cellular senescence, its activity can be detected using a chromogenic assay using 5-bromo-4-chloro-3-indoyl β-D-galactopyranoside (X-gal), which converts to an insoluble blue compound. Therefore, many researchers have utilized the SA-β-galactosidase assay as an indicator of cellular senescence (G.Untergasser et al., Experimental Gerontology, 2003 & F Debacq-Chainiaux et al., Nature protocols, 2009).
- It is suggested to add separate subsection containing the explanations of all applied abbreviations.
We have described the separate subsection for abbreviations in the line 351-360.
“Abbreviations
DMEM; Dulbecco's modified Eagle's medium
ECM; Extracellular matrix
FBS; Fetal bovine serum
HDFs; human dermal fibroblasts
PBS; Phosphate-buffered saline
SA-β-galactosidase; Senescence-associated beta galactosidase
TMSCs; human tonsil-derived mesenchymal stem cells
TMSC-NVs; Nanovesicle derived from human tonsil-derived mesenchymal stem cells
UV; Ultraviolet B”
- Paper should be written in a third person, not in a first one.
We have revised 2. Material and methods and 3. Results in a third person.
- Quality of Figures should be improved. Furthermore, some figures should be enlarged because they are poorly legible (e.g. Figure 1b or 1d etc.).
We have replaced the figure to the enlarged one.
- From editorial viewpoint, references in brackets should be placed at the end of the sentence before the dot – this should be corrected through the whole article.
We have placed the references in brackets at the end of the sentence before the dot.
- Final conclusions should be improved to be more specified and understandable. Now, this section consists of two sentences wherein the first is very, very long. It is better to list the highlights from the investigations in few, specified sentences.
We have revised the conclusion as short sentences.
“This study demonstrated that a human tonsil derived mesenchymal stem cells-derived nanovesicles share characteristics of the exosome and increase the proliferation of senescent HDFs. The senescence-associated β-galactosidase activities and vinculin expression in senescent cells were reduced by treatment of TMSC-NVs. The gene expression of extracellular matrix production and anti-oxidant gene were enhanced by treatment of TMSC-NVs. These findings could contribute to the development of skin rejuvenation tools and desirable cosmetic products once the clinical examination shows the promising effectiveness.”
- Section References should be improved to be consistent. Now, some references contain the whole journal title and in the case of some of them their abbreviations have been given.
We have confirmed and revised the references.
- Article should be re-checked grammatically.
We have grammatically confirmed the manuscript.

Round 2
Reviewer 1 Report
The revised version of this paper is well written. The authors have addressed all concerns.